# Effectiveness of blocking primers and a peptide nucleic acid (PNA) clamp for 18S metabarcoding dietary analysis of herbivorous fish

Chiho Homma[1], Daiki Inokuchi[2], Yohei Nakamura[2], Wilfredo H. Uy[3], Kouhei Ohnishi[1,2], Haruo Yamaguchi[1,2], Masao Adachi[1,2]*

1 The United Graduate School of Agricultural Sciences, Ehime University, Matsuyama, Japan, 2 Faculty of Agriculture and Marine Science, Kochi University, Nankoku, Kochi, Japan, 3 Institute of Fisheries Research and Development, Mindanao State University at Naawan, Naawan, Misamis Oriental, Philippines

* madachi@kochi-u.ac.jp

**Data Availability Statement:** All relevant data are within the paper and its Supporting Information files.

## Abstract

The structure of food webs and carbon flow in aquatic ecosystems can be better understood by studying contributing factors such as the diets of herbivorous fish. Metabarcoding using a high-throughput sequencer has recently been used to clarify prey organisms of various fish except herbivorous fish. Since sequences of predator fish have dominated in sequences obtained by metabarcoding, we investigated a method for suppressing the amplification of fish DNA by using a blocking primer or peptide nucleic acid (PNA) clamp to determine the prey organisms of herbivorous fish. We designed three blocking primers and one PNA clamp that anneal to fish-specific sequences and examined how efficient they were in suppressing DNA amplification in various herbivorous fish. The results showed that the PNA clamp completely suppressed fish DNA amplification, and one of the blocking primers suppressed fish DNA amplification but less efficiently than the PNA clamp. Finally, we conducted metabarcoding using mock community samples as templates to determine whether the blocking primer or the PNA clamp was effective in suppressing fish DNA amplification. The results showed that the PNA clamp suppressed 99.3%–99.9% of fish DNA amplification, whereas the blocking primer suppressed 3.3%–32.9%. Therefore, we propose the application of the PNA clamp for clarifying the prey organisms and food preferences of various herbivorous fish.

## Introduction

Herbivorous fish play an important role in determining the biological structure of shallow-water environments and carbon flow in aquatic ecosystems [1, 2]. The decline in abundance of herbivorous fish in tropical coral reefs causes a phase shift from coral- to algal-dominated reefs [3, 4], whereas an increase in fish herbivory in temperate reefs leads to a shift from algal forests to deforested barrens [5, 6]. Selective feeding by herbivorous fish affects algae and benthic-

**Funding:** The author(s) received no specific funding for this work.

**Competing interests:** The authors have declared that no competing interests exist.

invertebrate species composition [7, 8]. A meta-analysis on the fate of photosynthetic carbon in marine ecosystems estimates that 30%–40% of the micro- and macroalgae net primary production of photosynthetic carbon is channeled to heterotrophs through herbivory [9]. Therefore, the structure of food webs and the functional roles of herbivorous fish in aquatic ecosystems can be elucidated by assessing the food preferences and diets of herbivorous fish.

Several methods have been used to investigate the food habits of herbivorous fish: multiple-choice experiments [10, 11], direct feeding observations [12], stable isotope ratio analysis [13], and stomach/gut content analysis [14–16]. Multiple-choice experiments and direct feeding observations are practical for large and browsing herbivorous fish that feed on macroalgae but not for small or grazing herbivorous fish that feed on microalgae. Stable isotope analysis provides information on taxonomic groups of prey but is not applicable for identifying the prey species. Stomach content analysis has been the most widely used method, but the fragmentation of prey makes detailed taxonomical identification difficult [17], especially in herbivorous fish with pharyngeal teeth such as parrotfish [15]. DNA barcoding of organisms in the gut contents of *Siganus fuscescens* [18] and barcoding on the feeding substrates of *Scarus ovifrons* [19] have been conducted to determine the prey organisms of herbivorous fish. The DNA barcoding [18, 19] accompanied by single-stranded conformational polymorphism (SSCP) analysis or molecular cloning of PCR products circumvents the problem described above, but these approaches need DNA sequencing of many PCR products from the SSCP analysis or DNA sequencing of many clones derived from the PCR products, which are laborious and time-consuming [20].

Recently, the prey organisms of various carnivores in Perciformes, Squamata, and Passeriformes have been clarified via metabarcoding using a high-throughput sequencer (HTS) and universal primers for various eukaryotic organisms [21], but the method has not been applied to determine the prey organisms of herbivorous fish. Since universal primers in metabarcoding analysis sometimes amplify sequences derived from the predators as well as their prey organisms [22], it is critical to suppress the polymerase chain reaction (PCR) amplification of predator DNA in the analysis. Blocking primers and peptide nucleic acid (PNA) clamps have been used to suppress the DNA amplification of various predators [23]. Blocking primers are modified primers with an added C3 spacer at the 3′ end, which suppresses DNA amplification in PCR. Vestheim and Jarman [24] reported the development of blocking primers that bind specifically to suppress the sequence of krill (*Euphausia superb*), thus revealing the prey. Furthermore, blocking primers have been applied in metabarcoding to clarify the prey organisms of various fish [25–32], however they have not been applied to herbivorous fish.

PNAs are synthetic nucleic acids of nucleo-bases that are linked to a peptide backbone [33]. When they bind to complementary DNA, they inhibit polymerase elongation and PCR amplification [34]. PNA clamps have also been applied to metabarcoding to clarify fish diet [35, 36], although there have been fewer reports on PNA clamps than on blocking primers. Terahara et al. [37] developed PNA clamps that specifically bind to Japanese eel (*Anguilla japonica*) sequences and reported that they successfully suppressed the DNA amplification of predatory fish and identified their prey organisms. However, whether blocking primers or PNA clamps are suitable than blockers for metabarcoding fish prey organisms has not been examined.

In this study, to apply the blockers to metabarcoding to clarify the gut contents of herbivorous fish, we investigated designing blockers that bind specifically to fish. The optimal annealing temperature and concentration of blockers for suppressing the PCR amplification of fish DNA were determined. Next, we conducted metabarcoding using mock community samples as templates and determined whether the blocking primer or the PNA clamp was effective in suppressing fish DNA amplification and clarifying the gut contents of herbivorous fish.

## Materials and methods

### Ethic statement

Although experiments on fish do not require any legal procedures or permission in Japan, in order to avoid causing pain to the specimens, we followed all applicable institutional and/or national guidelines for the care and use of animals (mammals, birds, and reptiles) in this study; i.e., Act on Welfare and Management of Animals (Notice of the Ministry of the Environment No. 105 of October 1, 1973), Standards relating to the Care and Keeping and Reducing Pain of Laboratory Animals (Notice of the Ministry of the Environment No. 88 of 2006), Fundamental Guidelines for Proper Conduct of Animal Experiment and Related Activities in Academic Research Institutions under the jurisdiction of the Ministry of Education (Notice of Ministry of Education No. 71, 2006), and Guidelines for Proper Conduct of Animal Experiments (established by the Science Council of Japan on June 1, 2006). Fish specimens in the Philippines were collected under the Gratuitous Permit no. 0148–18 issued by the Department of Agriculture-Bureau of Fisheries and Aquatic Resources and the auspices of the research collaboration between Mindanao State University and Kochi University. No special collection permit is required for Japanese fish samples because they are not regulated or endangered species. Samples from Muroto and Kutsu were obtained from fishermen, and those from Yokonami were collected at the water in front of the Yokonami Rinkai Experimental Station (Kochi University). We got verbal consent of all participants in this study.

### Fish samples

Samples of 24 species from various taxonomic groups (4 orders, 9 families, and 19 genera) were collected to investigate the applicability of blockers to marine and freshwater herbivorous fish (S1 Table). Fish samples were collected by bait fishing, spear fishing, seine net, and hand net. Details of sampling are described in S1 Table. No endangered or protected species were involved in the field of sampling. Fish samples collected by spear fishing or bait fishing were euthanized by rapid severance of the head from the spinal at the time of sampling. Small fishes were euthanized by rapid chilling using ice-chilled water. The fish samples were sent to the Laboratory of Aquatic Environmental Science (Kochi University), where the species were identified by morphological characters [38] and then frozen at −25°C. Herbivory generally involves a fish diet consisting of at least 25%−50% plants [39]. Thus, not all species in this study were strictly herbivorous; some were like omnivorous (*Cyprinus carpio*, *Pomacentrus coelestis*), or possibly target protein-rich autotrophic microorganisms (parrotfish) [40]. Dietary information was obtained from Froese and Pauly [41] and published dietary data [42, 43].

### Designing a universal reverse primer and blockers

To develop blocking primers and a PNA clamp that anneal to a partially overlapped region with the 3′ end of a eukaryotic-universal primer (Fig 1), we designed a new universal reverse primer that binds to a universal region among various eukaryotes and is adjacent to the fish-specific sequence (Fig 1). Multiple alignments were prepared using the 226 sequences of the 18S rDNA V8–V9 region of Teleostei, the 2,713 sequences of the 18S rDNA region of various taxa of eukaryotes (S2 Table), which were obtained from the National Center Biotechnology Information (NCBI), and the 24 fish sequences determined in this study described later using Geneious 104 11.1.5 (Biomatters, Auckland, New Zealand). The universal reverse primer, 18SV9R, which binds to all eukaryotic sequences of the 18S rDNA V9 region tested, was newly

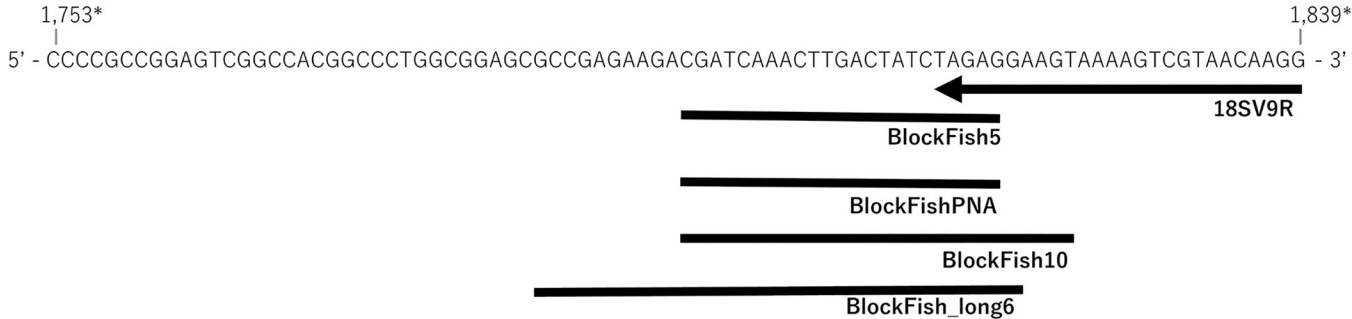

**Fig 1. Binding site of each blocking primer and PNA clamp used in this study.** The arrow indicates the binding site of the universal reverse primer, 18SV9R. The bars indicate the blocking primers or PNA clamp. *: Base position based on the 18S rDNA sequence of *Verasper variegatus* (Acc. No. EF126043).

designed (Table 1, Fig 1, and S2 Fig) and used in combination with the eukaryote universal primer, V8f [44] which binds to the 18S rDNA V8 region. The blocking primer and PNA clamp were designed in a region that overlap with 18SV9R and specific to Teleostei (S2 Fig).

To suppress PCR amplification of fish-derived 18S rDNA, we developed three blocking primers and one PNA clamp target fish-specific sequences of the 18S rDNA V8–V9 region, namely, BlockFish5, BlockFishPNA, BlockFish_long6, and BlockFish10 (Table 1), thereby annealing to a partially overlapped region with 5, 5, 6, and 10 bp of the universal reverse primer 18SV9R, respectively (Fig 1).

## Preparation of fish genomic DNA, PCR condition, and DNA sequencing

The dorsal fins or dorsal muscles of the fish samples were cut into sections (approximately 0.1 g each), which were transferred into separate 1.5 ml microcentrifuge tubes (Greiner Bio-One, Kremsmünster, Austria). The genomic DNA of the fish samples was extracted using a DNeasy Plant Mini Kit (Qiagen, Hilden, Germany) according to the manufacturer's protocol. The concentration of genomic DNA of each sample was quantified using a Qubit® 3.0 Fluorometer (Thermo Fisher Scientific, Waltham, MA, USA) according to the manufacturer's protocol. The 18S ribosomal RNA gene V8–V9 region of the fish samples was amplified by PCR using universal primer V8f [44] (Table 1) and 18SV9R (Table 1). The PCR mixture was as follows:

**Table 1. List of the primers and the PNA clamp used in this study for metabarcoding.**

| Name | Primer/PNA | Direction | Sequence (5'–3')*1 | Binding site*2 | Length | Tm*3 | Reference |
|---|---|---|---|---|---|---|---|
| V8f | Primer | Forward | ATAACAGGTCTGTGATGCCCT | 1,487–1,507 | 21 | 58.8 | [44] |
| 18SV9R | Primer | Reverse | CCTTGTTACGACTTYTMCTTCCTCTA | 1,814–1,839 | 26 | 58.6–61.5 | This study |
| V8f_tag | Primer | Forward | TCGTCGGCAGCGTCAGATGTGTATAAGA GACAGATAACAGGTCTGTGATGCCCT | 1,487–1,507 | 21 | 58.8 | [44] |
| 18SV9R_tag | Primer | Reverse | GTCTCGTGGGCTCGGAGATGTGTATAAG AGACAGCCTTGTTACGACTTYTMCTTCCTCTA | 1,814–1,839 | 26 | 58.6–61.5 | This study |
| BlockFish5 | Primer | Reverse | CTCTAGATAGTCAAGTTTGATCG | 1,796–1,818 | 23 | 54.2 | This study |
| BlockFish10 | Primer | Reverse | ACTTCCTCTAGATAGTCAAGTTTGATCG | 1,796–1,823 | 28 | 60.4 | This study |
| BlockFish_long6 | Primer | Reverse | CCTCTAGATAGTCAAGTTTGATCGTCTTCTCGGC | 1,786–1,819 | 34 | 67.4 | This study |
| BlockFishPNA | PNA | Reverse | CTCTAGATAGTCAAGTTTGATCG | 1,796–1,818 | 23 | 77.5 | This study |

*1: Underline: overhang adapter sequence

*2: Base position based on the 18S rDNA sequence of *Verasper variegatus* (Acc. No. EF126043)

*3: Tm calculated using the nearest neighbor method

18.56 μl water for injection (Nipro Pharma, Osaka, Japan), 1 μl genomic DNA of each fish (10 ng/μl), 2.5 μl of 10 × Ex *Taq* buffer (TaKaRa Bio, Shiga, Japan), 2.0 μl 2.5 mM dNTPmix (dATP, dTTP, dGTP, dCTP), 0.13 μl TaKaRa Ex *Taq* (5 units/μl; TaKaRa Bio), and 0.4 μl of each primer (12.5 μM) as described above. The PCR consisted of 30 cycles of 98˚C for 10 s, 60˚C for 30 s, and 72˚C for 1 min, then one cycle of 72˚C for 7 min. The PCR products were electrophoresed with ExcelBand 100 bp +3K DNA Ladder (SMOBiO Technology, Hsinchu, Taiwan) using 1.5% (w/v) agarose gel stained with ethidium bromide and viewed under an ultraviolet light [45]. Following the method of Hartley and Bowen [46], the products were purified to remove excess primers and dNTPs using 30% (w/v) polyethylene glycol (PEG) solution. Direct sequencing of the product to determine the 18S rDNA sequence of each fish sample was performed by using the primer V8f or 18SV9R with a BigDye Terminator v3.1 Cycle Sequencing Kit (Applied Biosystems Japan, Tokyo, Japan) following the manufacturer's protocol.

## Selection of blockers for suppressing PCR amplification of fish DNA

The usefulness of the blocking primers and the PNA clamp were evaluated by carrying out PCR in a total volume of 25 μl containing 22.5 μl of Platinum PCR SuperMix High Fidelity (Thermo Fisher Scientific), 0.2 μM of universal primers, 2.0 μM of BlockFish5, BlockFish10, BlockFish_long6, or BlockFishPNA, and 10 ng of genomic DNA of the herbivorous fish *Scarus ovifrons* (accession number of 18S rDNA: LC639933) as a template. As a control, sterile milli-Q water was added instead of the blocking primer or PNA clamp. PCR consisted of 30 cycles: denaturation at 94˚C for 15 s, annealing at 60˚C, 65˚C, and 66˚C for 30 s, and elongation at 68˚C for 1 min. To evaluate the blocking efficiency of each blocking primer and the PNA clamp, amplicons with or without the blocking primer or PNA clamp were run by gel electrophoresis with ExcelBand 100 bp +3K DNA Ladder (SMOBiO Technology) using 1.5% (w/v) agarose gel stained with ethidium bromide, which was viewed under an ultraviolet light. Photographs were taken with a camera (PowerShot G1 X Mark II, PC2049; Canon Inc., Tokyo, Japan), and the density of the bands was measured using ImageJ ver. 1.52i (National Institutes of Health, MD, USA). The relative amount of amplicon was calculated as follows:

$$\textbf{Relative amount of amplicon}~(\%) = [(\mathbf{x} - \mathbf{a})/(\mathbf{y} - \mathbf{a})] \times \mathbf{100}$$

where $x$ represents the band density of amplicon obtained by PCR with the blocking primer or PNA clamp; $y$ represents the band density of amplicon obtained by PCR without any blocker; and $a$ represents the background of the negative control obtained by PCR with sterilized milli-Q water instead of the template DNA.

Blocker suppression of fish DNA amplification was evaluated by the Tukey–Kramer test using R ver. 3.5.1.

## Optimization of PCR conditions for suppressing fish DNA amplification

To clarify the optimal concentration of the blocking primer and PNA clamp required for suppressing PCR amplification of fish DNA, the18S rDNA V8–V9 region was amplified with 0.2 μM of V8f and 18SV9R primers and various concentrations (0.2 μM, 1.0 μM, and 2.0 μM) of BlockFish_long6 and BlockFishPNA using *Scarus ovifrons* DNA as a template at annealing temperatures of 60˚C and 65˚C, respectively. The amplicons were run by gel electrophoresis, and the density of the bands was measured using the method described above and then compared using the Tukey–Kramer test.

## Applicability of the blocking primer and PNA clamp to various herbivorous fish

To investigate the applicability of BlockFish_long6 and BlockFishPNA to various herbivorous fish species, PCR was performed using the genomic DNA of 24 herbivorous fish (S1 Table) as templates. PCR was performed under optimal conditions to suppress the amplification of the *S. ovifrons* sequence as described above. BlockFish_long6 (2.0 μM) was added at an annealing temperature of 65˚C. BlockFishPNA (0.2 μM) was added at an annealing temperature of 60˚C. The amplicons were run by gel electrophoresis, and the density of the bands was measured using the method described above. The relative amount of amplicon was calculated as follows:

$$\text{Relative amount of amplicon without blocker } (\%) = [(x - a)/(y - a)] \times 100$$

where $x$ represents the band density of amplicon obtained by PCR without the blocking primer or PNA clamp; $y$ represents the band density of the 500 bp band of the 100 bp ladder marker; and $a$ represents the background of the negative control obtained by PCR with sterilized milli-Q water instead of the template DNA.

$$\text{Relative amount of amplicon with blocker } (\%) = [(x - a)/(y - a)] \times 100$$

where $x$ represents the band density of the amplicon obtained by PCR with the blocking primer or PNA clamp; $y$ represents the band density of amplicon obtained by PCR without any blocker; and $a$ represents the background of the negative control obtained by PCR with sterilized milli-Q water instead of the template DNA.

To evaluate the suppression of fish DNA amplification by using blockers, the Tukey–Kramer test was performed using R ver. 3.5.1.

## Mock community preparation and sequencing

To assess the efficiency of the blockers to suppress DNA amplification, three artificial samples of mock communities were prepared (Table 2). Eight organisms were selected to represent a wide range of genetically distinct eukaryotic taxa, as shown in Table 2. The three mock community samples selected were the fish *Scarus ovifrons*, three taxa of macroalgae, one taxon of microalgae, and three taxa of marine benthic animals. Each organism was identified based on the morphological characteristics and phylogenetic position, which was analyzed with the 18S rDNA sequences determined by the method described below.

The genomic DNA sequences of eight mock sample organisms were extracted using a Power-Soil® DNA Isolation Kit (Qiagen) according to the manufacturer's protocol. The amount of DNA in each sample was quantified using the Qubit® 3.0 Fluorometer (Thermo Fisher Scientific) according to the manufacturer's protocol. The 18S rDNA V8–V9 region of the eight organisms was amplified with V8f [44] and 18SV9R (Table 1) primers using the genomic DNA as templates under the conditions described above. PCR products were purified using a NucleoSpin PCR Clean-up Gel Extraction Kit (Macherey-Nagel, Düren, Germany) according to the manufacturer's protocol. Purified PCR products were cloned into the T vector pMD20 (TaKaRa Bio) according to the manufacturer's protocol. The vectors were introduced into *Escherichia coli* using a NEBuilder® HiFi DNA Assembly Cloning Kit (New England Biolabs, MA, USA) following the manufacturer's protocol. The plasmids were extracted from the transformed bacteria using the Fast Gene Plasmid Mini Kit (Nippon Genetics, Tokyo, Japan). After extraction, the 18S rDNA sequence that was inserted into the vectors was determined using the primer V8f with a BigDye Terminator v3.1 Cycle Sequencing Kit (Applied Biosystems Japan) following the manufacturer's protocol.

Plasmid DNA (10 μg) containing the 18S rDNA V8–V9 region of each organism was digested with 180 units of *Hind*III (18 units/μl; Toyobo, Osaka, Japan) according to the

manufacturer's protocol. The digested plasmid DNA was purified using a NucleoSpin PCR Clean-up Gel Extraction Kit (Macherey-Nagel), following the manufacturer's protocol. The concentration of the digested plasmids was quantified using a Qubit® 3.0 Fluorometer (Thermo Fisher Scientific) following the manufacturer's protocol. The purified plasmids were mixed in the ratios shown in Table 2.

To evaluate the suppression of fish 18S rDNA amplification with blockers during metabarcoding, the 18S rDNA V8–V9 region was duplicately amplified via PCR with V8f_tag and 18SV9R_tag (Table 1) and the blocker using the mock community samples as templates (Table 3) under the condition described above. DNA amplification in the two samples was determined by gel electrophoresis. Duplicate PCR amplicons were combined and purified using AMPure XP beads (Beckman Coulter, CA, USA) following the manufacturer's instructions. Next, Dual-Index PCR to add the index to the amplicons was performed according to the "16S Metagenomic Sequencing Library Preparation" protocol (Illumina, CA, USA) using the Nextera® XT Index Kit v2 Set A (Illumina). The PCR products were purified using AMPure XP beads (Beckman Coulter). The amount of DNA in each sample was quantified using a Qubit® 3.0 Fluorometer (Thermo Fisher Scientific) according to the manufacturer's protocol. The purified amplicons were diluted to 4 nM with 10 mM Tris-HCl (pH 8.5). The diluted amplicons of 24 samples were mixed in a single 1.5 ml tube. The mixture (5 μl) was subjected to MiSeq Reagent Nano Kit v2 (2 × 250 pair end; llumina) on an in-house MiSeq platform (Illumina).

## Processing and analysis of data

Sequence data obtained by MiSeq were used to make pair–end sequences in Mothur ver. 1.36.0 [47] on Galaxy 1.39.5.0 (URL: https://usegalaxy.org/). Paired-end reads were processed

**Table 2. Details of the experimental design and the organisms used to create the three mock community samples (Mock1–3) used for the metabarcoding.**

| Organism | Taxon | Location | Date | Acc. No. of 18S rDNA | Mock1[*1] | Mock2[*1] | Mock3[*1] |
|---|---|---|---|---|---|---|---|
| *Scarus ovifrons* | Teleostei | Muroto Misaki, Muroto City, Kochi, Japan (33.266, 134.160) | 2015. 07. 13 | LC639933 | 1 | 10 | 100 |
| *Zonaria diesingiana* | Ochrophyta | Otsuki, Hata, Kochi, Japan (32.463, 132.433) | 2019. 08. 27 | LC639948 | 1 | 1 | 1 |
| *Gelidium* sp. | Rhodophyta | Otsuki, Hata, Kochi, Japan (32.463, 132.433) | 2019. 08. 27 | LC639950 | 1 | 1 | 1 |
| *Ulva reticulata* | Chlorophyta | Otsuki, Hata, Kochi, Japan (32.463, 132.433) | 2019. 08. 27 | LC639946 | 1 | 1 | 1 |
| *Symbiodinium* sp. | Dinoflagellata | Pet shop[*2] | 2019. 11. 13 | LC639945 | 1 | 1 | 1 |
| *Pagurus filholi* | Arthropoda | Tei, Konan City, Kochi, Japan (33.311, 133.451) | 2019. 10. 04 | LC639951 | 1 | 1 | 1 |
| Nereididae sp. | Annelida | Tei, Konan, City, Kochi, Japan (33.311, 133.451) | 2019. 10. 04 | LC639947 | 1 | 1 | 1 |
| Euphylliidae sp. | Cnidaria | Pet shop[*1] | 2019. 11. 13 | LC639949 | 1 | 1 | 1 |

[*1]: Number in column showed a ratio of each organism in the mock community samples.

[*2]: *Symbiodinium* sp. DNA was obtained from a Euphylliidae sp. purchased from a pet shop located in Kochi City, Kochi, Japan.

**Table 3. Samples used for evaluating the suppression of fish 18S rDNA amplification with and without blockers using the mock community samples as templates.**

| Sample No. | Blocker | Annealing temp. (˚C) | Template[*1] |
|:---:|:---:|:---:|:---:|
| 1 | - | 65 | Mock1 |
| 2 | - | 65 | Mock2 |
| 3 | - | 65 | Mock3 |
| 4 | BlockFish_long6 | 65 | Mock1 |
| 5 | BlockFish_long6 | 65 | Mock2 |
| 6 | BlockFish_long6 | 65 | Mock3 |
| 7 | - | 60 | Mock1 |
| 8 | - | 60 | Mock2 |
| 9 | - | 60 | Mock3 |
| 10 | BlockFishPNA | 60 | Mock1 |
| 11 | BlockFishPNA | 60 | Mock2 |
| 12 | BlockFishPNA | 60 | Mock3 |

[*1]: Mock samples described in Table 2.

using Mothur v.1.33.0. Primer sequences were removed (pdiffs = 3), and no ambiguous bases were allowed. The maximum homopolymer size was 8 bp, and short sequences below 270 bp were removed. The remaining sequences were de-replicated and screened for chimeras using UCHIME [48]. Reads were clustered into operational taxonomic units (OTU) using preclustering with 98% similarly. BLAST was conducted using NCBI local blast + ver. 2.7.1 with 90% similarity.

Suppression of fish DNA amplification was calculated as follows:

$$\text{Suppression } (\%) = (b/c) \times 100$$

where $b$ represents the proportion of reads of fish (Teleostei) to the total reads obtained for each sample with the blocker; and $c$ represents the proportion of reads of fish (Teleostei) to the total reads obtained for each sample without the blocker.

# Results

## Selection of blockers and optimization of PCR conditions for suppressing fish DNA amplification

To select blockers that are useful for suppressing fish DNA amplification, PCR was performed at annealing temperatures of 60˚C, 65˚C, and 66˚C using four blockers targeting the 18S rDNA V9 region (Table 1, Fig 1). The results showed that the relative amount of amplicons with BlockFish5 and BlockFish10 at the annealing temperatures of 60˚C and 65˚C were not significantly different from those that did not use blockers as the controls (Fig 2A and 2B). BlockFish5 and BlockFish10 significantly reduced the relative amount of amplicons by approximately 50% from that of control at 66˚C (Fig 2C, Tukey–Kramer test: $p < 0.01$). BlockFish_long6 significantly reduced the relative amount of amplicons by 7.8% and 23.7% from those of the controls at 65˚C and 66˚C, respectively (Tukey–Kramer test: $p < 0.01$), whereas the blocker did not significantly reduce the relative amount of amplicons at 60˚C (Fig 2). In contrast to these blocking primers, BlockFishPNA reduced the relative amount of amplicons by 100% from those of the controls at all the temperatures tested (Fig 2, Tukey–Kramer test: $p < 0.01$). When PCR was conducted without any blocker, the amount of amplicons obtained at the annealing temperature of 66˚C was significantly lower than that at 60˚C (S1 Fig, Tukey–

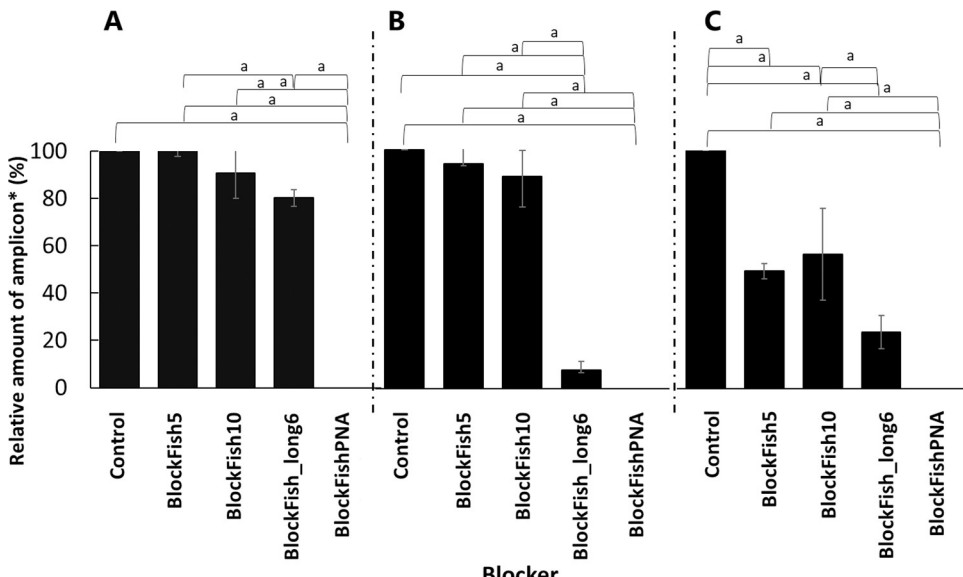

**Fig 2. Suppression of fish 18S rDNA amplification with the blocking primers and PNA clamp.** Annealing temperatures: A: 60˚C; B: 65˚C; C: 66˚C. Control: 18S rDNA amplification without the blocking primer or PNA clamp. [a]: Tukey–Kramer test: $p < 0.01$. *: Relative amount of amplicon when the amount of amplicon of the control was considered to be 100%.

Kramer test: $p < 0.01$). BlockFish_long6 and BlockFishPNA were further investigated at the annealing temperatures of 65˚C and 60˚C, respectively.

The concentrations of the blocking primer and the PNA clamp that are suitable for suppressing fish DNA amplification were evaluated via PCR conducted with various concentrations (0.2 µM, 1.0 µM, and 2.0 µM) of BlockFish_long6 and BlockFishPNA. The results showed that the higher the concentration of BlockFish_long6, the lower the relative amount of amplicons (Fig 3A, Tukey–Kramer test: $p < 0.01$). BlockFishPNA showed 100% suppression of fish DNA amplification at all concentrations tested (Fig 3B). Considering these results, 2.0 µM (10 times the concentration of universal primers) of BlockFish_long6 and 0.2 µM (the same concentration as that of the universal primers) of BlockFishPNA were further investigated.

### Applicability of the blockers to various herbivorous fish

To investigate the applicability of BlockFish_long6 and BlockFishPNA to various herbivorous fish species, PCR was performed using blockers with the genomic DNA of 24 herbivorous fish (S1 Table) employed as templates. When BlockFish_long6 was used, the relative amount of amplicon was 0% for *Cyprinus carpio*, *Carassius auratus langsdorfii*, and *Chlorurus bleekeri*, i.e., the suppression efficiency was 100%; the relative amount of amplicon was < 50% for the 16 fish species showing ≧50% suppression efficiency; and the relative amount of amplicon was ≧50% for four species showing suppression efficiency of less than 50% (Table 4). In using the genomic DNA of *Naso vlamingii* as the template, 18S rDNA was not amplified at the annealing temperature of 65˚C, even when the blocking primer was not added (Table 4). In contrast, BlockFishPNA effectively suppressed amplification of all tested fish DNA (Table 4).

### Mock community analysis

To evaluate the applicability of the blockers to metabarcoding analysis, the 18S rDNA V8–V9 region was amplified via PCR with blockers using the mock community samples (Table 3) as

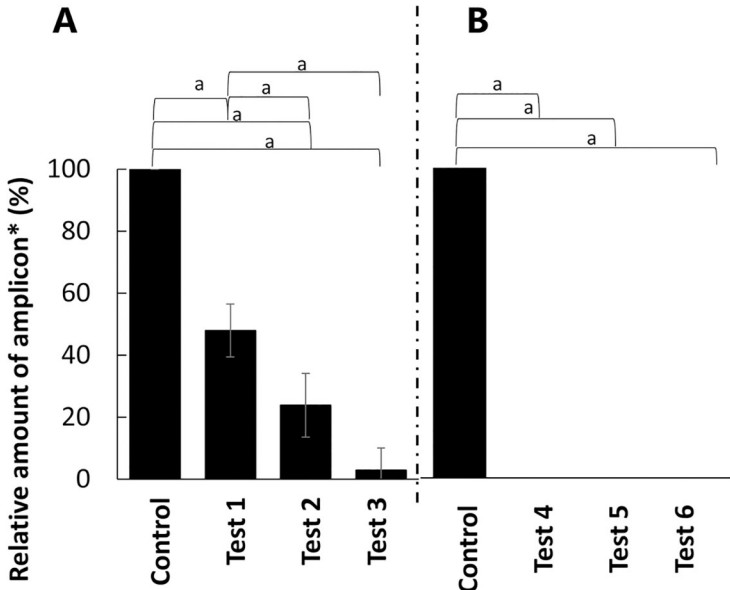

**Fig 3. Effect of the concentration of the blocking primer or PNA clamp in a PCR cocktail on the suppression of fish 18S rDNA amplification.** A: BlockFish_long6 added; B: BlockFishPNA added. Tests 1, 4: 0.2 μM added; Tests 2, 5: 1.0 μM added; Tests 3, 6: 2.0 μM added. Control: 18S rDNA amplification without the blocking primer or PNA clamp. [a]: Tukey–Kramer test: $p < 0.01$. [*]: Relative amount of amplicon when the amount of amplicon of the control was considered to be 100%.

templates. The results showed that the proportions of fish reads of samples 1, 2, and 3 without blockers at an annealing temperature of 65˚C were, 18.4%, 74.2%, and 97.0%, respectively (Fig 4). The proportions of fish reads in samples 4, 5, and 6 at an annealing temperature of 65˚C with BlockFish_long6 were 12.3%, 64.1%, and 93.8%, respectively (Fig 4). Compared with each control (PCR amplified without blocker), the suppression efficiencies of fish reads by Block-Fish_long6 in the samples were 32.9%, 13.6%, and 3.3%, respectively.

Thus, it was deduced that a higher amount of fish DNA in the mock community was core-lated with less effective suppression of fish DNA with BlockFish_long6 (Fig 4). The proportions of fish reads in samples 7, 8, and 9 without blockers at an annealing temperature of 60˚C were 19.3%, 77.5%, and 97.3%, respectively. In contrast, the proportions of fish reads in samples 10, 11, and 12 at an annealing temperature of 60˚C with BlockFishPNA were 0.00%, 0.2%, and 0.7%, respectively (Fig 4).

Compared with each control (PCR amplified without blocker), the suppression efficiencies of fish reads by BlockFishPNA in the samples were 99.9%, 99.7%, and 99.3%, respectively. These results showed that the suppression efficacy of fish DNA amplification with Block-FishPNA did not decrease, even when the concentration of fish DNA added as a template was increased, whereas that with BlockFish_long6 decreased when the concentration of fish DNA increased.

## Discussion

### Design of blockers suitable for suppressing fish DNA amplification

Both BlockFish5 and BlockFish10, which have 5 bp- and 10 bp-overlap regions in the blocking primers with the reverse universal primer, respectively, hardly suppressed fish DNA

**Table 4. PCR amplification of herbivorous fish 18S rDNA and suppression of the amplification with the blockers.**

| Species | Relative amount of amplicon[*1] | | Relative amount of amplicon[*2] | |
|---|---|---|---|---|
| | No blocker 65°C | No blocker 60°C | BlockFish_long6 65°C | BlockFishPNA 60°C |
| *Cyprinus carpio* | ++ | +++ | - | - |
| *Carassius auratus langsdorfii* | + | +++ | - | - |
| *Amblygobius phalaena* | +++ | +++ | + | - |
| *Chrysiptera cyanea* | +++ | +++ | ++ | - |
| *Pomacentrus coelestis* | +++ | +++ | + | - |
| *Rhabdoblennius nitidus* | ++ | ++ | + | - |
| *Petroscirtes variabilis* | +++ | ++ | + | - |
| *Ellochelon vaigiensis* | +++ | +++ | + | - |
| *Calotomus spinidens* | +++ | +++ | ++ | - |
| *Calotomus japonicus* | ++ | +++ | ++ | - |
| *Cetoscarus ocellatus* | +++ | +++ | + | - |
| *Scarus psittacus* | ++ | +++ | + | - |
| *Scarus ovifrons* | ++ | +++ | + | - |
| *Scarus ghobban* | +++ | +++ | + | - |
| *Chlorurus bleekeri* | + | +++ | - | - |
| *Naso vlamingii* | - | + | ND | - |
| *Prionurus scalprum* | +++ | +++ | + | - |
| *Zebrasoma scopas* | +++ | +++ | + | - |
| *Acanthurus nigrofuscus* | +++ | +++ | + | - |
| *Acanthurus dussumieri* | +++ | +++ | ++ | - |
| *Ctenochaetus striatus* | +++ | +++ | + | - |
| *Siganus argenteus* | +++ | +++ | + | - |
| *Girella punctata* | +++ | +++ | + | - |
| *Girella leonina* | +++ | ++ | + | - |

[*1]: Relative amount of amplicon when the amount of 500 bp band of the 100 bp ladder marker was regarded as 100%.

[*2]: Relative amount of amplicon when the amount of amplicon of control without blocker was regarded as 100%. -: amplicon was not observed. +: 0%< relative amount <50%,++: 50%≦ relative amount <100%, +++: 100%≦ relative amount. ND: no data.

amplification at annealing temperatures of 60°C and 65°C, and the relative amounts of amplicons with BlockFish5 and BlockFish10 were not significantly different at any of the annealing temperatures tested. The results indicated that the extension of the overlap regions in the blockers from 5 bp to 10 bp with the universal primer did not enhance the suppression of fish DNA amplification. In contrast, BlockFish_long6 suppressed fish DNA amplification at higher efficiencies than those of BlockFish5 and BlockFish10. The reason the relative amount of amplicons with the former blocker was smaller than those with the latter was reasoned to be due to the more efficient annealing of the former than the latter to the target fish DNA as a result of the high Tm of the former (67.4°C) compared with that of the latter (54.2°C and 60.4°C). These results suggest that the length of the overlap region in a blocking primer with the PCR primer may not be important, and a high Tm of a blocking primer may be a key factor in suppressing DNA amplification.

## Annealing temperature suitable for suppressing fish DNA amplification

Suppression of fish DNA amplification increased as the annealing temperature of the blocking primer increased. In particular, BlockFish_long6 suppressed DNA amplification more

efficiently at 65°C and 66°C compared with other blocking primers, whereas the relative amount of the products amplified with only the universal primers decreased at higher annealing temperatures, especially at 66°C. Thus, 65°C, at which DNA amplification was highly suppressed with the blocker and a relatively high amount of amplicons was obtained without the blocker, was selected as the optimal annealing temperature for the blocking primer as well as for the universal primers used. The results suggest that it is necessary to clarify the annealing temperatures that are critical to maintaining a certain level of DNA amplification efficiency so that target DNA amplification is efficiently suppressed.

Because BlockFishPNA completely suppressed fish DNA amplification at all of the annealing temperatures tested (60°C, 65°C, and 66°C), 60°C was selected as the annealing temperature for BlockFishPNA, at which the highest amount of amplicon was obtained without a blocker. Suppression of DNA amplification at all temperatures was considered to have occurred due to the Tm (77.5°C) of BlockFishPNA being >15°C higher than those of the universal primers (58.8°C and 58.6°C –61.5°C). Chow et al. [49] reported that a PNA clamp with 73.7°C Tm targeting the Japanese spiny lobster (*Panulirus japonicus*) suppressed DNA amplification at all of the tested annealing/extension temperatures (53°C, 55°C, 58°C, and 60°C), which were >13°C lower than the Tm of the PNA clamp. These results suggest that the PNA clamp has a high efficiency for suppression of DNA amplification, even under low annealing temperatures. Consequently, PNA clamps might be more applicable than blocking primers for suppressing target DNA amplification.

## Concentration of blockers suitable for suppressing fish DNA amplification

In this study, the higher the concentration of the blocking primers, the more efficient the suppression of the target DNA amplification. Su et al. [30] investigated the effect of various concentrations of blocking primers on the suppression of fish DNA (*Acanthopagrus latus*, *Pampus argenteus*, and *Scomberomorus commerson*) amplification. They added the blocking primer at increasing concentrations of 1-, 3-, 5-, and 10-fold that of the universal primers and found that the blocker efficiency in suppressing DNA amplification of these fish correspondingly increased [30]. Other studies have demonstrated that the higher the concentration of the blocking primer, the more efficient the suppression of the target DNA amplification [24, 50, 51]. These results suggest that the optimal concentration of a blocking primer is 10-fold that of the universal primers.

In terms of the concentration of the PNA clamp, Chow et al. [49] added PNA at approximately 5- and 10-fold higher concentrations that that of the universal primers and found that both concentrations were highly effective in suppressing DNA amplification. In the present study, DNA amplification was completely suppressed, even if the PNA clamp was added at the same concentration as that of the universal primers. These results suggest that the PNA clamp may be used at the same concentration as that of the PCR primers. Considering that a PNA clamp is expensive, we recommend clarifying the minimum concentration required of each PNA clamp to stably suppress DNA amplification.

## Applicability of the blockers to various herbivorous fish

When BlockFishPNA was used to suppress the amplification of DNA of various herbivorous fish, the relative amount of amplicon was 0% for all fish species tested, i.e., 100% suppression efficiency of PCR amplification. In contrast, the relative amount of amplicons with Block-Fish_long6 differed depending on the genomic DNA used as templates, even if all the fish genomic DNA had no mismatch with the BlockFish_long6 sequence, which indicated that the difference in the relative amount of amplicons with BlockFish_long6 was not caused by the

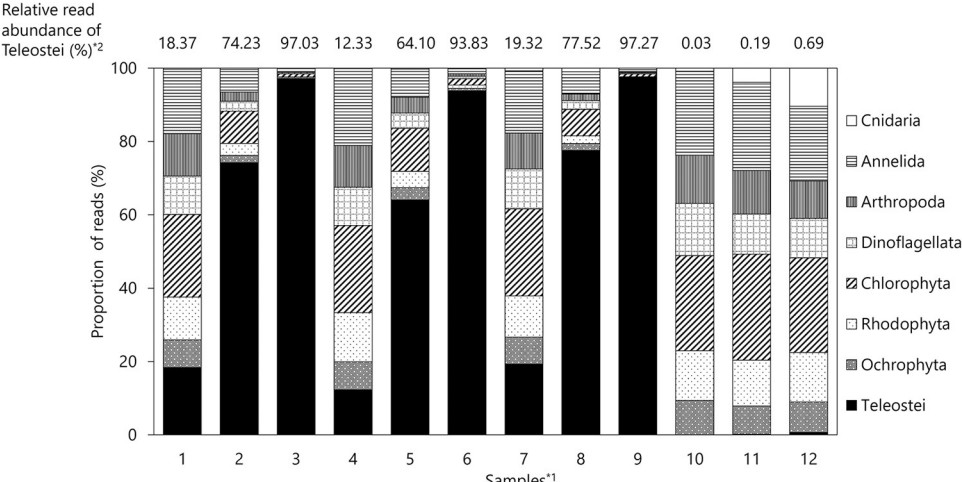

**Fig 4. Proportion of read numbers of the organisms contained in the mock community samples obtained by metabarcoding analysis.** [*1]: Samples shown in Table 3. [*2]: Proportion of fish reads to all reads obtained for each sample.

presence of mismatches. The reason why the amount of amplicons of the samples used differed may be because the balance between DNA amplification and suppression by the blocker might be altered by differences in DNA amplification efficiencies due to the extent of 'contaminants' in the template genomic DNA. This hypothesis suggests that purifying the template DNA thoroughly is a beneficial step when a blocking primer is used.

## Effect of the blockers on suppressing fish DNA amplification in the metabarcoding of mock communities

In the results of metabarcoding of the mock community samples, the suppression efficiency of BlockFish_long6 was low (3.3%–32.9%, Fig 4), whereas the suppression efficiency of BlockFish_long6 was high (92.18%) when PCR was performed using only fish genomic DNA as a template under the same conditions (Fig 2B). In relation to these results, Takahashi et al. [31] designed a blocking primer targeting carnivorous fish (Lutjanidae) and investigated how efficient it was in suppressing the amplification of target DNA. They reported that the efficiency of suppressing DNA amplification was high when only Lutjanidae (target) DNA was used as a template, whereas the efficiency was low when a mock sample containing not only the target DNA but other DNA were used as templates. The reason for the difference in the results was not described in their report. Considering their results as well as our results, suppression efficiencies of the blocking primer differed with the results of the PCR test using only the target DNA as the template and the mock test containing other DNA as well as target DNA may be because 'contaminants' in some DNA added to the mock samples might affect DNA amplification efficiencies and alter the balance between DNA amplification and suppression. This hypothesis suggests that purifying the template DNA used to prepare mock samples is beneficial.

On the other hand, the suppression efficiency of BlockFishPNA was high when metabarcoding of mock community samples and PCR were performed using only the fish genomic DNA as a template. The reason why the PNA clamp was more efficient than the blocking primer was considered to be due to the Tm of BlockFishPNA being higher than that of BlockFish_long6. The results suggest that a PNA clamp is more effective than a blocking primer to

consistently suppress the amplification of the target DNA when metabarcoding is conducted using a blocker.

The relative proportion of reads of non-fish organisms in the mock samples with and without BlockFish_long6 was almost the same, which indicated that the BlockFish_long6 developed in this study did not bind to the DNA of non-fish organisms. The fact that there were ≥5 bp mismatches between the sequences of the organisms other than fish (*Scarus ovifrons*) in the mock community, and the sequence of BlockFish_long6 suggests that the blocking primers do not generally bind to sequences of organisms with ≥5 bp mismatches (S3 Table). On the other hand, Piñol et al. [51] developed a blocking primer targeting *Orius majusculus* (Insecta) and investigated the usefulness of the blocking primer when using mock community samples for metabarcoding. They reported that the blocking primer suppressed DNA amplification, even when the DNA had 5 bp mismatches with the sequence of the blocking primer was used as a template in the mock community sample. These results suggest that investigating the 'blocking characteristic' of each blocking primer is desirable.

The effect of a PNA clamp on suppressing the amplification of target and non-target DNA in mock community samples has not been investigated. As far as we know, this study is the first to report the usefulness of a PNA clamp when performing metabarcoding of a mock community. In this study, the relative proportion of reads of non-fish organisms in the mock samples with and without BlockFishPNA was almost the same, which indicated that the BlockFishPNA designed in this study did not suppress the DNA amplification of sequences of organisms having at least 5 bp mismatches with the sequence of the PNA clamp and did not affect the proportion of reads of non-target organisms. Moreover, Terahara et al. [37] developed a PNA clamp targeting *Anguilla japonica* (Teleostei) and reported that the PNA clamp did not suppress DNA amplification when the template DNA had 2 or 3 bp mismatches with the sequence of the PNA clamp. These results indicate that the PNA clamp did not suppress DNA amplification of the sequences of organisms having at least 5 bp or potentially 2 bp mismatches with the sequence of the PNA clamp, and the PNA clamp did not affect the proportion of reads of non-target organisms.

## Conclusion

In this study, we designed several DNA blockers to suppress fish DNA amplification during metabarcoding of the gut contents of herbivorous fish and investigated the suppression efficiency of these blockers. Among them, BlockFish_long6, a blocking primer designed in this study, showed high suppression efficiency of fish DNA amplification when only fish DNA was used as a template. However, when BlockFish_long6 was used for metabarcoding the mock community, the suppression efficiency of fish DNA amplification was low. On the other hand, BlockFishPNA, the PNA clamp designed in this study, showed a high suppression efficiency of fish DNA amplification, even when the mock samples were used as a template for metabarcoding. Therefore, BlockFishPNA is considered to be optimal for metabarcoding of prey organisms of herbivorous fish and is expected to be applied to detect and analyze the prey organisms of various herbivorous fish species.

## Supporting information

**S1 Fig. Effect of annealing temperature in the control (18S rDNA amplification without the blocking primer or the PNA clamp) on fish 18S rDNA amplification.** a: Tukey-Kramer test: $p < 0.01$. *: Relative amount of amplicon when the amount of amplicon at 60°C was regarded as 100%.
(TIF)

**S2 Fig. Multiple alignment prepared for designing the universal primer and the blockers.**
(TIF)

**S1 Table. List of herbivorous fish used in this study.**
(DOCX)

**S2 Table. Number of 18S rDNA sequences of each taxon taken from NCBI for designing the universal reverse primer and fish (Teleostei) blockers.**
(DOCX)

**S3 Table. Number of mismatches between the sequence of each blocker and the 18S rDNA sequence of each taxon in the region of the blocker.**
(DOCX)

## Acknowledgments

We thank Shingo Seki (Kochi University), Anabelle Dece A. Espadero (Kochi University), Zenzo Imoto (Kochi University), and Kenzo Yamagata, for collecting fish samples. We also thank Chetan Chandrakant Gaonkar (Kochi University) and Hiroshi Funaki (Ehime University) for their support with the data analysis of the sequence. We also thank Dennis Murphy (Ehime University) for the revising this manuscript linguistically.

## Author Contributions

**Conceptualization:** Chiho Homma, Daiki Inokuchi, Yohei Nakamura, Masao Adachi.

**Data curation:** Chiho Homma, Kouhei Ohnishi, Haruo Yamaguchi, Masao Adachi.

**Formal analysis:** Masao Adachi.

**Investigation:** Chiho Homma, Daiki Inokuchi, Kouhei Ohnishi, Masao Adachi.

**Methodology:** Chiho Homma.

**Resources:** Yohei Nakamura, Wilfredo H. Uy, Masao Adachi.

**Supervision:** Masao Adachi.

**Validation:** Chiho Homma, Masao Adachi.

**Visualization:** Chiho Homma, Masao Adachi.

**Writing – original draft:** Chiho Homma, Masao Adachi.

**Writing – review & editing:** Yohei Nakamura, Masao Adachi.

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
