## [Decision Letter · Decision Letter 0]

17 Feb 2022

PONE-D-21-29551Effectiveness of blocking primers and a peptide nucleic acid (PNA) clamp for 18S metabarcoding dietary analysis of herbivorous fishPLOS ONE

Dear Dr. Masao Adachi, 

Thank you for submitting your manuscript to PLOS ONE. After careful consideration, we feel that it has merit but does not fully meet PLOS ONE’s publication criteria as it currently stands. Therefore, we invite you to submit a revised version of the manuscript that addresses the points raised during the review process.

Please submit your revised manuscript by March 21, 2022. f you will need more time than this to complete your revisions, please reply to this message or contact the journal office at plosone@plos.org. Please include the following items when submitting your revised manuscript:A rebuttal letter that responds to each point raised by the academic editor and reviewer(s). You should upload this letter as a separate file labeled 'Response to Reviewers'.A marked-up copy of your manuscript that highlights changes made to the original version. You should upload this as a separate file labeled 'Revised Manuscript with Track Changes'.An unmarked version of your revised paper without tracked changes. You should upload this as a separate file labeled 'Manuscript'.If applicable, we recommend that you deposit your laboratory protocols in protocols.io to enhance the reproducibility of your results. Protocols.io assigns your protocol its own identifier (DOI) so that it can be cited independently in the future. For instructions see: https://journals.plos.org/plosone/s/submission-guidelines#loc-laboratory-protocols. Additionally, PLOS ONE offers an option for publishing peer-reviewed Lab Protocol articles, which describe protocols hosted on protocols.io. Read more information on sharing protocols at https://plos.org/protocols?utm_medium=editorial-email&utm_source=authorletters&utm_campaign=protocols.

We look forward to receiving your revised manuscript.

Kind regards,

Ram Kumar, Ph.D.

Academic Editor

PLOS ONE

" ext-link-type="uri" xlink:type="simple">https://journals.plos.org/plosone/s/file?id=ba62/PLOSOne_formatting_sample_title_authors_affiliations.pdf"

 “CH No. 4350 The Japan Science Society https://www.jss.or.jp/en/

NO”

Additional Editor Comments:

The manuscript have been reviewed by the three independent reviewers.

I thank you for considering the Plos One as potential vehicle of your research outcomes.

I am pleased to inform that all the three reviewers have recommended publication of the manuscript after minor revision. The comments and suggestions given by all the three reviewers are very pertinent . I would request you to revise the manuscript accordingly and resubmit for final decision. In addition to comments by three reviewers I think that the details of designing and selecting the primers are ambiguous , reviewers have also commented on this issue.

The manuscript is acceptable after incorporating all amendments suggested by the reviewers.

Reviewers' comments:

Reviewer's Responses to Questions

**Comments to the Author**

1. Is the manuscript technically sound, and do the data support the conclusions?

Reviewer #1: Yes

Reviewer #2: Yes

Reviewer #3: Yes

2. Has the statistical analysis been performed appropriately and rigorously? 

Reviewer #1: Yes

Reviewer #2: Yes

Reviewer #3: Yes

3. Have the authors made all data underlying the findings in their manuscript fully available?

Reviewer #1: Yes

Reviewer #2: Yes

Reviewer #3: Yes

4. Is the manuscript presented in an intelligible fashion and written in standard English?

Reviewer #1: Yes

Reviewer #2: Yes

Reviewer #3: Yes

5. Review Comments to the Author

Reviewer #1: The study aims to assess the efficiency of PNA clamp to suppress the amplification of fish DNA and compare it with the efficiency of blocking primers. 24 species of herbivorous fish are tested in the study. The authors design three blocking primers and one PNA clamp to determine the organisms that are a part of the diet of herbivorous fish. The results clearly indicate the dominance of PNA clamp over blocking primers in suppressing the fish DNA amplification by a huge margin.

The results from this study could be of a good application in studying aquatic food web and nutrient cycle while being species specific. The manuscript is well written and comprehensible. All my comments and questions are stated herewith.

The authors have stated the limitations of previously used methods, but the limitations of DNA barcoding accompanied with molecular cloning as a method could be elaborated a little more to consolidate their point of the need for a new method (inclusion of PNA clamp in this case during amplification) because the point that it is laborious and time consuming (line 44-47) could be applicable to most of the processes described and is general in my opinion. As DNA barcoding has been previously used for determining the prey organism of Siganus fuscescens and Scarus ovifrons, I think it would be better to specify the limitations from that experiment as well to strengthen the claims made by the author.

Regarding the table no. 4. For langsdorfii, no observations are included in the table and the genus name has not been provided, please address to that as it has been mentioned in line no. 325-327 of the manuscript in relation to a 100% suppression by BlockFish_long7

Also, Since Naso vlamingii does not seem to produce amplicons without blockers at 65 and at 60 degrees Celsius, and produces only 0-50% of relative amount of amplicons, was it tested at lower annealing temperatures than 60 degrees to see if more amplicons are being produced for Naso vlamingii without the blocker. Was there any limitation that did not allow more observations for the said species such as the sample size or availability.

Please include the page numbers and make sure to review the submission guidelines for the journal to check the formatting of the manuscript.

A general question for someone not a specialist in this field, This is more of a personal curiosity regarding the use of V9 and V8f region, Is there a possibility to use any more regions on 18S rDNA or an inclusion of more regions of 18S rDNA as biomarker, if yes would that be beneficial or the results would be similar in preparing the blockers and PNA clamp?

Reviewer #2: The manuscript shows how efficiently the gut microbiology of herbivorous fish can be investigated by focusing on 18S rDNA of only prey organisms and blocking the DNA content of fish. The study was novel as the prey content has not been identified in herbivorous fish. The manuscript is well written. Would suggest minor revision. The changes required are:

1. The details of designing and selecting the primers were not clear in the manuscript. If added in supplementary, I would suggest to add that in the materials and methods section where the primer designing has been mentioned. Also mention from which organism/s each primers has been designed.

2. I feel metabarcoding results should also be properly mentioned. The prey organisms mentioned in Table 2 were only used for conducting mock tests? Incorporate the metabarcode analysis of prey organisms (the list of organisms or classes of organisms that are observed in fish by PCR blocking technique).

The manuscript can be accepted after incorporating these minor corrections.

Reviewer #3: Few corrections:

1. BlockFish_long7 has only 6 nucleotide overlap with the 18SV9R which does match with the text in line 119.

2. In the Table 2 labels of columns Mock 1, Mock 2 and Mock 3 does not give the idea about what the numbers given in the columns represent (in line 236 mentioned but it should be clearly labeled in table also)

6. PLOS authors have the option to publish the peer review history of their article (what does this mean?). If published, this will include your full peer review and any attached files.

Reviewer #1: **Yes: **Malayaj Rai

Reviewer #2: No

Reviewer #3: No

---

## [Author Response · Author response to Decision Letter 0]

23 Feb 2022

Dear Reviewers,

We are grateful for your valuable comments and suggestions that have helped to improve our manuscript (MS), entitled "Effectiveness of the blocking primer and PNA clamp against for 18S metabarcoding dietary analysis of herbivorous fishes" by Chiho Homma, Daiki Inokuchi, Yohei Nakamura, Wilfredo H. Uy, Kouhei Ohnishi, Haruo Yamaguchi, Masao Adachi as an Original Research Article for publication in PLoS One. We have considered the comments and suggestions in the revised version of our MS. Responses to your comments are presented below.

Responses to Reviewer #1’s Comments:

Comment 1: The authors have stated the limitations of previously used methods, but the limitations of DNA barcoding accompanied with molecular cloning as a method could be elaborated a little more to consolidate their point of the need for a new method (inclusion of PNA clamp in this case during amplification) because the point that it is laborious and time consuming (line 44-47) could be applicable to most of the processes described and is general in my opinion. As DNA barcoding has been previously used for determining the prey organism of Siganus fuscescens and Scarus ovifrons, I think it would be better to specify the limitations from that experiment as well to strengthen the claims made by the author.

Response 1: Thank you for your suggestion. Considering your suggestion, we described the limitations from the DNA barcoding in the revised Introduction of our manuscript (revised MS Page 3, Lines 45 – 49) as described below:

‘The DNA barcoding [18,19] accompanied by single-stranded conformational polymorphism (SSCP) analysis or molecular cloning of PCR products circumvents the problem described above, but these approaches need DNA sequencing of many PCR products from the SSCP analysis or DNA sequencing of many clones derived from the PCR products, which are laborious and time-consuming [20].’

Comment 2: Regarding the table no. 4. For langsdorfii, no observations are included in the table and the genus name has not been provided, please address to that as it has been mentioned in line no. 325-327 of the manuscript in relation to a 100% suppression by BlockFish_long7

Response 2: Carassius auratus langsdorfi is one species name. In the original manuscript, there were separated into two lines, which made it difficult to understand. Therefore, the species name was shown in a single line in Table 4 in the revised MS. 

Comment 3: Also, Since Naso vlamingii does not seem to produce amplicons without blockers at 65 and at 60 degrees Celsius, and produces only 0-50% of relative amount of amplicons, was it tested at lower annealing temperatures than 60 degrees to see if more amplicons are being produced for Naso vlamingii without the blocker. Was there any limitation that did not allow more observations for the said species such as the sample size or availability.

Response 3: In Table 4, we showed effectiveness of BlockFish_long6 and BlockFishPNA to various fishes. No amplicon of Naso vlamingii was obtained at 65℃, but an amplicon of it was obtained at 60°C, although the amount of the amplicon was small (0-50%). We think that it is not necessary to test the effectiveness of blockers at lower annealing temperatures than 60 degrees, because non-specific amplification may occur if the annealing temperature is lowered below this temperature (60 degrees) which is about Tm of the universal primer. So we did not test temperatures less than 60℃. 

Comment 4: Please include the page numbers and make sure to review the submission guidelines for the journal to check the formatting of the manuscript.

Response 4: Following your suggestion, we have added the page numbers in the revised MS. Thank you.

Comment 5: A general question for someone not a specialist in this field, This is more of a personal curiosity regarding the use of V9 and V8f region, Is there a possibility to use any more regions on 18S rDNA or an inclusion of more regions of 18S rDNA as biomarker, if yes would that be beneficial or the results would be similar in preparing the blockers and PNA clamp?

Response 5: We think that it is possible to use regions other than 18S rDNA V8-V9 used in this study, such as mitochondrial CO1 as barcode regions. In the cases, it is important to consider a use of a PNA clamp or a use of a blocking primer having higher Tm than that of a universal PCR primer. In addition, it is desirable to optimize the concentration of the blocker and the annealing temperature as investigated in this study. 

Responses to Reviewer #2’s Comments:

Reviewer #2: The manuscript shows how efficiently the gut microbiology of herbivorous fish can be investigated by focusing on 18S rDNA of only prey organisms and blocking the DNA content of fish. The study was novel as the prey content has not been identified in herbivorous fish. The manuscript is well written. Would suggest minor revision. The changes required are:

Comment 1: The details of designing and selecting the primers were not clear in the manuscript. If added in supplementary, I would suggest to add that in the materials and methods section where the primer designing has been mentioned. Also mention from which organism/s each primers has been designed.

 Response 1: We created a multiple alignment using the 226 Teleostei sequences, 2,713 sequences of various eukaryotic organisms other than Telostei from NCBI (Table S2), and the 24 fish sequences determined in this study. In the revised manuscript, we showed the alignment in Fig. S2 (Please check the figure in an attached file as Response to reviewers) which shows the annealing sites of the newly designed universal reverse primer 18SV9R, the blocking primers and the PNA clamp used in this study. 

Comment 2: I feel metabarcoding results should also be properly mentioned. The prey organisms mentioned in Table 2 were only used for conducting mock tests? Incorporate the metabarcode analysis of prey organisms (the list of organisms or classes of organisms that are observed in fish by PCR blocking technique).

Response 2: We are sorry that we have not yet applied the metabarcode analysis using the blockers to analysis of prey organisms of herbivorous fishes. In future, we would like to report prey organisms of various herbivorous fishes analyzed by a metabarcode analysis using the blockers reported in this study. 

Responses to Reviewer #3’s Comments:

Comment 1: BlockFish_long7 has only 6 nucleotide overlap with the 18SV9R which does match with the text in line 119.

Response 1: Considering your comment, we have changed the name of the blocker from BlockFish_long7 to BlockFish_long6 in the text as well as all figures and tables. Thank you.

Comment 2: In the Table 2 labels of columns Mock 1, Mock 2 and Mock 3 does not give the idea about what the numbers given in the columns represent (in line 236 mentioned but it should be clearly labeled in table also)

Response 2: Considering your comment. we have explained the meaning of the number in the footnote of the table as shown below:

‘Number in column showed a ratio of each organism in the mock community samples.’

References

 We have inserted spaces before volume numbers and deleted spaces before page numbers. 

 We have added information about article numbers or a volume number of some references (Ref.: 5, 19, 26, 27, 29, 31, 32, 36, 37).

We have changed the doi of Ref: 44.

---

## [Decision Letter · Decision Letter 1]

18 Mar 2022

Effectiveness of blocking primers and a peptide nucleic acid (PNA) clamp for 18S metabarcoding dietary analysis of herbivorous fish

PONE-D-21-29551R1

Dear Dr. adachi,

We’re pleased to inform you that your manuscript has been judged scientifically suitable for publication and will be formally accepted for publication once it meets all outstanding technical requirements.

Kind regards,

Ram Kumar, Ph.D.

Academic Editor

PLOS ONE

Additional Editor Comments (optional):

Dear Prof Masao Adachi

We’re pleased to inform you that your manuscript has been judged scientifically suitable for publication and will be formally accepted for publication once it meets all outstanding technical requirements.

Thank you ones again for considering the PLOSE ONE as potential vehicle of disseminating your research outcomes.

Prof. Ram Kumar

Handling Editor

Reviewers' comments:

Reviewer's Responses to Questions

**Comments to the Author**

1. If the authors have adequately addressed your comments raised in a previous round of review and you feel that this manuscript is now acceptable for publication, you may indicate that here to bypass the “Comments to the Author” section, enter your conflict of interest statement in the “Confidential to Editor” section, and submit your "Accept" recommendation.

Reviewer #1: All comments have been addressed

Reviewer #2: All comments have been addressed

2. Is the manuscript technically sound, and do the data support the conclusions?

Reviewer #1: Yes

Reviewer #2: Yes

3. Has the statistical analysis been performed appropriately and rigorously? 

Reviewer #1: Yes

Reviewer #2: Yes

4. Have the authors made all data underlying the findings in their manuscript fully available?

Reviewer #1: Yes

Reviewer #2: Yes

5. Is the manuscript presented in an intelligible fashion and written in standard English?

Reviewer #1: Yes

Reviewer #2: Yes

6. Review Comments to the Author

Reviewer #1: The study effectively shows the potential for use of blocking primer and 18S metabarcoding as a means of dietary analysis of herbivorous fish by blocking the expression of fish DNA during amplification. The study is novel and has potential for future works related to the accurate identification of prey using metabarcoding as well which would provide information that could be experimented in aquaculture industry. All my questions and suggestions have been addressed to.

Reviewer #2: All the questions have been addressed. In the acknowledgement section, pls remove 'the' mentioned in the line 502-503 (for "the" revising). Can be accepted.

7. PLOS authors have the option to publish the peer review history of their article (what does this mean?). If published, this will include your full peer review and any attached files.

Reviewer #1: **Yes: **Rai, Malayaj

Reviewer #2: No

---

## [Editor Report · Acceptance letter]

8 Apr 2022

PONE-D-21-29551R1 

Effectiveness of blocking primers and a peptide nucleic acid (PNA) clamp for 18S metabarcoding dietary analysis of herbivorous fish 

Dear Dr. Adachi:

I'm pleased to inform you that your manuscript has been deemed suitable for publication in PLOS ONE. Congratulations! Your manuscript is now with our production department. 

Kind regards, 

on behalf of

Professor Ram Kumar 

Academic Editor

PLOS ONE